# Duplication, Loss, and Evolutionary Features of Specific UDP-Glucuronosyltransferase Genes in Carnivora (Mammalia, Laurasiatheria)

**DOI:** 10.3390/ani12212954

**Published:** 2022-10-27

**Authors:** Mitsuki Kondo, Yoshinori Ikenaka, Shouta M. M. Nakayama, Yusuke K. Kawai, Mayumi Ishizuka

**Affiliations:** 1Laboratory of Toxicology, Department of Environmental Veterinary Science, Faculty of Veterinary Medicine, Hokkaido University, N18, W9, Kita-ku, Sapporo 060-0818, Japan; 2Water Research Group, Unit for Environmental Sciences and Management, North-West University, Private Bag X6001, Potchefstroom 2520, South Africa; 3Translational Research Unit, Veterinary Teaching Hospital, Faculty of Veterinary Medicine, Hokkaido University, Sapporo 060-0818, Japan; 4One Health Research Center, Hokkaido University, Sapporo 060-0818, Japan; 5Biomedical Sciences Department, School of Veterinary Medicine, The University of Zambia, P.O. Box 32379, Lusaka 10101, Zambia; 6Laboratory of Toxicology, Department of Veterinary Medicine, Obihiro University of Agriculture and Veterinary Medicine, Obihiro 080-8555, Japan

**Keywords:** wildlife, xenobiotic metabolism, in silico analysis, genome database, phase II metabolism, glucuronidation

## Abstract

**Simple Summary:**

In this study, we clarified the evolutional features of the UGT gene family in Carnivora. We firstly analyzed the gene synteny of UGT1As, 2Bs, ana 2Es and further demonstrated the phylogenetic analysis to reveal the evolutional gene duplication and loss event in Carnivora. We found specific UGT1A duplication in Canidae, brown bear and black bear, and UGT2Bs duplication in Canidae, some Mustelidae, and Ursidae. In addition, we observed gene contraction of UGT1A7–12 in Phocidae, Otariidae, and Felidae. This study strongly suggested closely related Carnivorans also showed significant evolutional differences of UGTs, and further imply the importance of appropriate approaches to assess pharmacokinetics and toxicokinetic from experimental animals.

**Abstract:**

UDP-glucuronosyltransferases (UGTs) are one of the most important enzymes for xenobiotic metabolism or detoxification. Through duplication and loss of genes, mammals evolved the species-specific variety of UGT isoforms. Among mammals, Carnivora is one of the orders that includes various carnivorous species, yet there is huge variation of food habitat. Recently, lower activity of UGT1A and 2B were shown in Felidae and pinnipeds, suggesting evolutional loss of these isoforms. However, comprehensive analysis for genetic or evolutional features are still missing. This study was conducted to reveal evolutional history of UGTs in Carnivoran species. We found specific gene expansion of UGT1As in Canidae, brown bear and black bear. We also found similar genetic duplication in UGT2Bs in Canidae, and some Mustelidae and Ursidae. In addition, we discovered contraction or complete loss of UGT1A7–12 in phocids, some otariids, felids, and some Mustelids. These studies indicate that even closely related species have completely different evolution of UGTs and further imply the difficulty of extrapolation of the pharmacokinetics and toxicokinetic result of experimental animals into wildlife carnivorans.

## 1. Introduction

UDP-glucuronosyltransferases (UGTs) are a superfamily of enzymes which catalyze the glucuronide conjugation reaction to both endogenous (e.g., bilirubin, bile acids, several hormones, and neurotransmitters) [1,2,3,4] and exogenous chemicals (e.g., prescribed drugs, veterinary drugs, plant-derived chemicals, and environmental pollutants) [5,6,7,8,9]. Using UDP-glucuronide as a donor, UGTs transfer a glucuronide moiety into substrate substances to increase hydrophilicity and generally drive deactivation for excretion through bile or urine. Hence, UGTs are generally considered among the major detoxification enzymes for mammals.

The mammalian UGTs superfamily consists of two families (1 and 2) and is subdivided into the 1A, 2A, and 2B subfamilies based on amino acid sequence levels [10,11]. A vast variety of mammal UGTs has diverged in each species through gene duplication and loss events [12,13]. For instance, in humans, there are 19 isoforms (9 in UGT1 and 10 in UGT2), showing different substrate specificities in enriching the metabolism with a wide range of chemicals. 

The mammalian order Carnivora includes a great diversity of families, such as canids (e.g., dogs and foxes), ursids (bears), mustelids (e.g., stoats and badgers), pinnipeds (e.g., seals and walruses), felids (e.g., small and large cats), hyaenids (hyaenas), herpestids (meerkat and mongooses), and viverrids (e.g., civets and genets) [14,15]. Almost all species are highly carnivorous. Due to this diet, almost all Carnivora are generally recognized as apex predators in their respective food chain. This high position in the food chain often entails strong effects of biomagnification and bioaccumulation of numerous environmental pollutants; numerous studies have indicated significant concentrations of persistent organic pollutants (POPs) in several Carnivorans. 

Recent studies on UGTs suggested strong genetic differences among Carnivora. Shrestha et al. (2011) [16] and Kakehi et al. (2015) [17] found a genetic dysfunction (pseudogene) in the major phenol-metabolizing enzyme UGT1A6 in Felidae, brown hyena (Parahyaena brunnea), and Otariidae. The carnivora-specific isoform UGT2B31 pseudogene also appeared to be present in all Felidae, indicating a limited capacity for glucuronidation [18]. Very limited numbers of UGT1A/2B isoforms were observed in these species, suggesting contraction of these genes during evolution [17,18]. 

These reports strongly suggest the presence of a large genetic diversity in UGT even within Carnivora; however, no further Carnivora species have yet been examined in this regard. Recent improvements in WGS (whole genome sequencing) techniques allow us to utilize a large volume of genomic data from a variety of wild carnivorans [19]. These data enable the comprehensive investigation of evolutionary history, including gene duplication/loss events, and sequence comparisons of each isoform. In the present study, we utilized genomic data from a large variety of Carnivora species to evaluate the genetic synteny of each UGT. We also conducted phylogenetic analysis of each subfamily to analyze evolutionary inter-species differences in this enzyme superfamily.

## 2. Materials and Methods

### 2.1. UGT Phylogenetic Analysis and UGT Gene Counts

Phylogenetic analyses were performed on the UGT genes of human (*Homo sapiens*), rat (*Rattus norvegicus*), mouse (*Mus musculus*), dog (*Canis lupus familiaris*), red fox (*Vulpes vulpes*), domestic ferret *(Mustela putorius furo*), ermine (*Mustela erminea*), mink (*Neovison vison*), Badger (*Meles meles*), North American river otter (*Lontra canadensis*), Eurasian river otter (*Lutra lutra*), sea otter (*Enhydra lutris kenyoni*), polar bear (*Ursus maritimus*), giant panda (*Ailuropoda melanoleuca*), black bear (*Ursus americanus*), brown bear (*Ursus arctos*), meerkat (*Suricata suricatta*), striped hyena (*Hyaena hyaena*), Domestic cat (*Felis catus*), Amur tiger (Panthera tigris), cheetah (*Acinonyx jubatus*), puma (*Puma concolor*), Canada lynx (*Lynx canadensis*), leopard (*Panthera pardus*), Lion (*Panthera leo*), Leopard cat (*Prionailurus bengalensis*), Weddell seal (*Leptonychotes weddellii*), harbor seal (*Phoca vitulina*), gray seal (*Halichoerus grypus*), Hawaiian monk seal (*Neomonachus schauinslandi*), northern elephant seal (Mirounga angustirostris), southern elephant seal (*Mirounga eonine*), northern fur seal (*Callorhinus ursinus*), Steller’s sea lion (*Eumetopias jubatus*), California sea lion (*Zalophus californianus*), and Pacific walrus (*Odobenus rosmarus divergens*). Sequences were retrieved by National Center for Biotechnology Information (NCBI) BLAST searches, using the following query sequences: for UGT1, human and dog UGT1A1, UGT1A6, UGT1A2, and UGT1A7; for UGT2, dog UGT2B31, human UGT2B8, cat UGT2E1, dog UGT2A1, and human UGT2A1. BLAST searches were conducted in the Nucleotide collection database (nr/nt) for each species using Blastn (optimized for somewhat similar sequences). This blast search for UGT1As were comprehensive enough to detect UGT1 and -2s in Carnivora, and UGT2Bs search also covered UGT1As in Carnivora. The gene sequences used are listed in Appendix A. For UGT1As, only the first exons for each isoform were analyzed, and for UGT2Bs, UGT2A1/2 and 2A3, all protein coding sequences were used. We also used BLAST to align query UGT sequences onto genome assembly sequences (RefSeq genome Database (refseq_genomes)), optimized for somewhat similar sequences) to make sure we did not miss other possible UGT isoforms on each species. The protein coding region of each isozyme was analyzed. The deduced amino acid sequences were aligned using MUSCLE (Multiple Sequence Comparison by Log-Expectation) and were used for model selection (minimal BIC) and construction of maximum likelihood trees (bootstrapping = 100) using MEGA X (Molecular Evolutionary Genetics Analysis) [20]. The JTT+G+I model was used. All positions containing gaps and missing data were eliminated, and alignment of the total length of the protein-coding sequence (1365 for the first exon of UGT1As and 1689 bp for UGT2s) was used for phylogenetic analysis. The results of phylogenetic analyses for human, mouse, rat, and dog UGT1A, UGT2A, and UGT2B genes were referenced to published phylogenic analyses [10,11] for verification. Lists of the food habitats of each Carnivora were referenced from publications and listed in Appendix A. 

### 2.2. Synteny Analysis of UGT Genes 

NCBI’s genome data viewer (https://www.ncbi.nlm.nih.gov/genome/gdv/: accessed on 1 September 2022) or JBrowse [21] were used to visualize the chromosomal synteny maps for each species, using freely available sequence data. The following latest genome assemblies were used: human Annotation Release 106, rat Annotation Release 105, mouse Annotation Release 105, dog Annotation Release 103, cat Annotation Release 102, Weddell seal Annotation Release 100, red fox Annotation Release 100, domestic ferret Annotation Release 101, ermine Annotation Release 100, American river otter Annotation Release 100, sea otter Annotation Release 100, polar bear Annotation Release 101, giant panda Annotation Release 103, brown bear Annotation Release 101, meerkat Annotation Release 100, striped hyena Annotation Release 100, Amur tiger Annotation Release 100, cheetah Annotation Release 101, puma Annotation Release 100, Canada lynx Annotation Release 102, leopard Annotation Release 100, harbor seal Annotation Release 100, gray seal Annotation Release 100, Hawaiian monk seal Annotation Release 100, northern fur seal Annotation Release 100, southern elephant seal Annotation Release 100, Steller’s sea lion Annotation Release 100, California sea lion Annotation Release 100, and Pacific walrus Annotation Release 101. UCSC (University of California, Santa Cruz, Santa Cruz, CA, US) BLAT (BLAST-like alignment tool) (http://genome.ucsc.edu/index.html accessed on 1 September 2022) was used for the additional confirmation of missing genes.

## 3. Results

### 3.1. UGT Synteny Analysis and Gene Counts

#### 3.1.1. UGT1A Coding Loci and Isoform Number in Mammals

UGT1A and UGT2A/B coding loci in rodents, humans, and carnivorans were analyzed and compared. The numbers of UGT isoforms in each Carnivora is shown in Figure 1. We counted the isoform numbers annotated on each assembly. UGT1A coding loci were highly conserved among Mammalia, in accordance with previous reports [12,22], and almost all isozymes were coded between MROH2A (maestro heat like repeat family member 2A) and USP40 (ubiquitin specific peptidase 40). Generally, UGT1As are coded by a common four exons (exons 2–5) and a unique alternative exon (exon 1), for which each gene product is spliced and named (UGT1A1–12). In this analysis, these features were also detected in all Carnivora analyzed (Figure 2). DNAJB3 (DnaJ Heat Shock Protein Family (Hsp40) Member B3) was located between exon 1 of UGT1A1 and UGT1A2, as in previous findings [10,11,17,22]. We also compared our isoform numbers with those annotated in published genomic data and found strong variation among Carnivora. In Canidae, more than 9 isoforms were detected (12 for dog, 10 for red fox and 9 for Arctic fox), followed by Ursidae (4–9 isoforms), Mustelidae (2–4 isoforms), pinnipeds (Odobenidae, Otariidae, and Phocidae) (1–3 isoforms), and Felidae (2) (Figure 1). We also compared length of coding locus as a means of tracking genetic duplication/loss events in these conserved regions (MROH2A to USP40) (Figure 1). However, there were substantial differences even within Ursidae. Polar bear and giant panda tended to have a limited number of isoforms and length of conserved region (polar bear: 4 isoforms, 98 kb; giant panda: 4 isoforms, 85.4 kb), while black and brown bear had relatively longer regions and number of isoforms (brown bear: 7 isoforms, 128.6 kb; black bear: 9 isoforms, 156.1 kb). Further BLAST search onto each genome sequence revealed other possible unannotated UGT1A7–12 isoforms, shown in Figure 1 and Figure 2, and included in further phylogenetic analysis. Manually curated unannotated isoforms are shown with “?” in Figure 2.

#### 3.1.2. UGT2A/B Coding Loci and Isoforms Number in Mammals

Similar to UGT1As, UGT2 coding loci were also conserved between SULT1B1 (sulfotransferase family 1B member 1) and YTHDC1 (YTH domain- containing protein 1) in all carnivora analyzed (Figure 3). This agrees with previous reports [10,11,18]. We again counted and compared the isoform number for UGT2A/2Bs and the respective lengths of coding loci (YTHDC1 to SULT1B1) (Figure 3). Similar to UGT1As, Canidae had a higher number of UGT2B isoforms (3–4 isoforms), while pinnipeds and Felidae had very limited numbers (0–1 isoforms) (Figure 1). Mustelidae had comparatively moderate numbers of UGT2Bs (1–4 isoforms), but with interspecies variation within the family. Blast analysis using UGT2B31 in dog as a query further revealed possible other isoforms in UGT2Bs in brown and black bear. We found nine further possible isoforms (eight complete and one partial) on unscaffolded contigs (NW_025929643 and NW_025929709) in brown bear and six other partial isoforms in black bear. Including these unscaffolded isoforms, brown bear had 11 UGT2B isoforms, which was highest number in the analyzed carnivora. Since some annotations contained incomplete UGT2B sequences, we attempted to avoid misestimation of gene counts by excluding shorter UGTs (<1500 bp) in Figure 3 (1590 bp for UGT2B7 in human) and distinguishing them as black boxes in Figure 2. 

### 3.2. Phylogenetic Analysis of UGTs

#### 3.2.1. Phylogenetic Analysis and Sequence Comparison of UGT1As in Carnivora

UGT1A phylogenetic analysis in this study revealed four possible clades for this subfamily in mammals: UGT1A1, UGT1A2–5, UGT1A6, and UGT1A7–12 (Figure 2). Almost all Carnivora had only a single orthologues gene of UGT1A1 and UGT1A6, suggesting strong conservation of these genes. However, we found that Canidae and brown/black bear showed specific expansions of gene isoform numbers: in canids, clade UGT1A2–5, and in both ursids and canids, clade UGT1A7–12 (Figure 4). Other Ursidae analyzed in this study (giant panda and polar bear) had no such specific duplication in these clades, and no other specific duplications in Carnivora UGT1As were observed in this study. In contrast, other carnivora families were found to have undergone genetic loss or contraction in the UGT1A7–12 clade. No species in Felidae and Phocidae had any annotated genes in this clade, suggesting complete loss of UGT1A7–12, and some species in Otariidae and Mustelidae (northern fur seal, California sea lion, ermine, river otter, and mink) had only one annotated isoform. 

#### 3.2.2. Phylogenetic Analysis of UGT2Bs in Carnivora

The phylogeny of the UGT2 family was also analyzed (Figure 5). We found three clades of UGT2s: UGT2As, UGT2Bs, and UGT2Es. Almost all UGT2E1s in Carnivora were registered as UGT2C1-like or UGT2A3-like in the NCBI database; in the cat, this isoform was recently renamed to UGT2E1 by the UGT nomenclature committee. Therefore, we renamed these isoforms to UGT2E1s based on their phylogeny, following the committee’s suggestions [10,11]. Almost all Carnivora, except for Felids, had paralogues of UGT2B31s within the same clade, which agrees with previously reported results [18]. We also observed some specific duplications and losses. Canidae, some Ursidae, and some Mustelidae had possible multiples of functional UGT2Bs, and these isoform duplications were clustered into species-specific clades, implying that the duplications were each species-specific. We also demonstrated the existence of annotated UGT2Es isoforms in all Felidae and Ursidae, and found isoforms annotated as “low-quality” UGT2C1-like in dog and red fox but in no other species. No other specific duplication or loss events were observed in the UGT2 family.

### 3.3. Sequence Comparison of UGT1As and 2Bs in Carnivora

Similar to the findings of previous studies, we also observed that all analyzed Felidae and Otariidae possessed UGT1A6 pseudogenes. We also found additional UGT1A6Ps in some species. In southern elephant seal, we could not find any annotated UGT1A6. Annotated UGT1A6 in sea otter contained a nonsense mutation and was registered as a pseudogene in the NCBI database.

We further compared the sequence of candidate UGT1A7–12 isoforms in northern fur seal, California sea lion, ermine, river otter, and mink, as these species only have a single isoform in this clade. We determined a specific common stop codon (TAG: 526–528 bp) in UGT1A7–12 in the two pinnipeds (Appendix A), indicating a dysfunction of these genes in the analyzed Otariidae. No possible nonsense mutations in UGT1A7–12 were observed in the three mustelids. 

We also compared the sequences of UGT2Bs, and found no dysfunctional mutation in any analyzed species except for UGT2B31 in all felids. 

## 4. Discussion

### 4.1. Relations between Diet and UGT2Bs Expansion

The generally accepted “animal-plant warfare” hypothesis considers the evolution of the xenobiotic metabolism as one of major defense mechanisms in animals against daily exposure to xenobiotics; in this regard, plant secondary metabolites are likely among the major sources of evolutionary pressure [13,24,25,26]. Several studies have shown that herbivorous mammals and birds have experienced a huge expansion of UGT families [27,28,29,30]. In placental mammals, Kawai et al. (2021) [31] recently demonstrated a relationship of herbivorous diet with a large number of UGT2B genes but not with UGT1A genes. This strongly suggests that UGT2Bs might be important for the daily metabolism of plant secondary metabolites. Moreover, a recent genomic analysis in woodrats (highly herbivorous) indicated significant duplication of UGT2Bs, in contrast to closely related omnivorous rats and deer mice [28]. Similarly, sika deer genomic analysis [29] suggested that genes in the UGT2B subfamily have a strong correlation with the adaptation of the species to a high-tannin diet. These reports underline the evolutionary importance of UGT2Bs in herbivorous adaptation. The present study demonstrated the expansion of UGT2Bs in Canidae (red fox and dog), brown bear, and some Mustelidae (badger, mink, and ermine) (Figure 1 and Figure 3). The canid and brown bear UGT2B expansion might be explained by the omnivorous diet of these species [32,33]. The brown bear showed a unique UGT2B expansion and the largest observed number of UGT2B isoforms; this was not the case for the closely related polar bear. Although data for such an inference are limited, black bear also showed a similar possibility for multiple functional UGT2Bs. Although these data of UGT2Bs expansions were still preliminary and further research to make sure these expansion and isoform number differences in these bears are needed, these data suggested the possible variation of UGT2Bs within Ursids. These two species have a very generalist diet, including much plant matter, such as green vegetation, fruits, cereals, and hard masts (e.g., nuts and acorns) [34,35,36,37,38]. It is likely that the observed UGT2B duplication is the result of adaptation to plant-based food items in the diet of these species.

In this analysis, results for the giant panda, a strictly herbivorous species, contradicted the “plant-animal warfare” assumption. UGT1As in this species indicated slightly contracted evolution compared to other omnivorous canids and ursids. The UGT2B gene family was lost completely. The cause for this might be the species’ exclusive bamboo diet. Since the giant panda has a diet of limited variety, they might take in much less variety of phytochemicals than other bears. This might lead them to have only limited number of UGT isoforms. Although bamboo itself also contains a certain variety of phytochemicals which may be metabolized by UGTs [39], the lower number of UGTs (which may be specialized for bamboo-phytochemical metabolism) might be enough for them to reach their daily intake of these chemicals. However, in mammals, some reports have shown the opposite traits, which show that specialists tend to have a wider variety of xenobiotic metabolism enzymes than generalists. In koala, to deal with their daily eucalypt-specialized diet, their genome seemed to expand CYP2C subfamilies [40]. Additionally, juniper-specialist woodrat and red tree voles, which have a Douglas fir-specialized diet, observed higher gene copy number of CYPs, compared to closely related generalist species [41,42]. However, these results have been mainly discussed with respect to CYP genes, which were not addressed in the current study. Seasonal fluctuations in gut microbiota have been suggested as an alternative strategy to provide a pathway for the metabolism of bamboo-related chemicals [43,44,45]. Further investigation about alternative pathways (e.g., sulfotransferase or other phase II metabolism enzymes) in giant panda to deal with these bamboo-derived phytochemicals or flavonoids have to be assessed in the future. 

Our phylogenetic analysis of UGT2Bs suggested dogs and foxes had totally different evolutional expansions. That suggests that UGT2Bs expansion in each species occurred after the divergence of these species. A previous report provided evidence that there were novel genomic adaptations among modern dogs in order to thrive on a diet rich in starch [46]. Together with our phylogenetic analysis, these reports suggested that UGT2Bs expansion might be specific to dogs, and not in wolves. Further genomic comparison between these species should be taken into consideration in future research. 

Interestingly, among canids, there were some differences in UGT2B evolution even in closely related species, such as red fox and arctic fox. We detected four annotated isoforms annotated in arctic fox, with three having a limited length of protein coding loci (XM_041724971.1: 1316 bp, XM_041724972.1: 842bp, XM_041726888.1: 183 bp). This information was automatically annotated based on NCBI annotation pipeline using RNA-seq data from several tissues. Further investigation is required to determine whether this species has a limited number of UGT2Bs, but the reduced length of 2B coding loci in this species in comparison to dog and red fox suggests it does. The dietary difference between arctic fox (an obligate carnivore) and red fox (a mesocarnivore) might also explain UGT2B contraction in the former [31,47,48,49], although a detailed examination is still limited.

Badgers also have an omnivorous diet, while American mink and ermine (both in the subfamily Mustelinae) are both highly carnivorous [50]. The UGT2B expansion observed in the latter two species does not agree with the patterns discussed above. This might suggest the presence of a possible opportunistic omnivorous diet in a common ancestor of Mustelinae, and UGT2B duplication and the current number of pseudogenes in these species could be the evolutionary footprint of that ancestor’s diet. Further research of other mustelids is required to clarify UGT2B evolution and the functional importance of UGT2B isoforms in this highly carnivorous family. 

### 4.2. UGT1A Evolution and Adaptation to Species-Specific Diets

In addition to UGT2Bs, UGT1As also showed significant expansion in Canidae and some Ursidae (brown bear and black bear) in this study. This also appears to be related to diet, as in UGT2Bs. A variable preference for plant food sources in Carnivora might partially explain the different UGT evolutional patterns and may have been a cause of the UGT1A expansion in adapting to species-specific plant diets. We also showed that the duplication of UGT1A2–5 in Canidae and UGT1A7–12 in Canidae and two bear species are family- or species-specific features, indicative of evolutionary events at these taxon levels. A previous study on avian UGTs suggested a correlation of herbivore diet and UGT1A numbers [30]. While the genetic expansion of avian UGT1As was observed especially in isoform clades relevant to mammalian UGT1A2–5, we observed expansion of both UGT1A2–5 and UGT1A7–12. Further investigation into the substrate specificity of each isoform and the relationship to species-specific dietary plants is needed to support this hypothesis. 

Recent genomic research in Hyaenidae also suggested UGT expansion in the aardwolf (*Proteles cristata*), an insectivorous species [51]. The authors discussed the possible role of this UGT expansion as a defense mechanism against termite toxins. However, they only detected expansion in orthologues gene of UGT2A1 (LOC480777), not in UGT2Bs. UGT2A1/2 has been known to specifically express in nasal epithelium and is regarded as playing significant roles in odor signal termination [52,53]; these enzymes are highly conserved among mammals. The observed UGT2A1 expansion in aardwolf and its connection with the detoxification of termite toxins is thus still unclear. Because genomic data availability is slightly limited, we could not extend our analysis to aardwolf and brown hyaena data. 

The Canidae/Ursidae-specific UGT1A expansion found in this study and the possible insectivore-derived expansion of UGTs in aardwolf indicate the importance of further research with a more comprehensive coverage of species and a more detailed partitioning of dietary habits (frugivore, folivore, nectarivore, insectivore, and others).

### 4.3. UGT Duplication/Loss and Relation to Functional Glucuronidation

The observed UGT1A and 2B duplication in Canidae found in this study strongly suggests a substantial capacity for and wide range of chemical acceptance for glucuronide conjugation in this clade. In a previous report, we revealed a strong glucuronidation capacity of in vitro dog liver microsome towards both UGT1A substrates [17] and UGT2B substrates [18]. Soars et al. (2001) [54] also reported a much stronger glucuronidation capacity for a wide range of chemicals of in vitro dog liver microsome compared to humans. This reasonably coincides with our results of genetic duplication in dogs, and our findings further indicate that these high capacities for glucuronidation may be present not only in dogs but also in other Canidae species, such as foxes. Still, our phylogenetic analysis suggested the duplication events in Canidae seemed to be species-independent, and further in vitro or in vivo analysis for foxes glucuronidation capacity is essential. We also observed strong contraction of UGT1As, especially the UGT1A6 pseudogene, UGT1A7–12 loss, and complete loss of UGT2Bs in all Felidae. These findings are in accordance with in vitro studies of limited capacity of UGTs for a wide range of chemicals in cats [55,56,57]. Similar features of in vitro limited glucuronidation were also observed in pinnipeds [17,18]. The present study adds further information on possible UGT1A7–12 loss in the entire Otariidae and Phocidae clades, although some Phocidae have intact UGT1A6 genes. Kakehi et al. [17] already revealed the limited number of UGT1A6–12 isoforms in Pacific walrus and further discussed the possible loss of UGT1A7–12 genes in pinnipeds as a cause of limited glucuronidation capacity in vitro. The present study’s results strongly support this hypothesis. UGT1A1–5s are generally considered as bilirubin-like-associated isoforms, whereas UGT1A6–12 isoforms are thought to be phenol-like-associated [58,59]. The present study thus suggests that almost all species of pinnipeds may have a more limited capacity of glucuronidation for a wide range of exogenous phenols than previously implied [17]. In vitro or in vivo activity of UGTs in Ursidae, Mustelidae, or any other Carnivora have not yet been studied, and further research is needed to understand the relationship between the genetics and function of UGTs in individual species. Genomic information for a wider range of Carnivora taxa (e.g., Ailuridae, Procyonidae, Mephitidae, Vivveridae, Nandiniidae, Prionodontidae, and Eupleridae) is also required to fill the gaps in the evolutionary history of UGT duplication/loss. 

We also detected slight differences of UGT2A3s and 2Es among Carnivora. We found the existence of UGT2A3 in some Canids and Ursids and UGT2Es in Felids and Ursids. UGT2A1 and 2A2s have been recognized as non-liver expressed isoforms, as mentioned above [52,53]. However, UGT2A3 were detected in liver of some primates and rodents [52,53,60,61], suggesting possible role for xenobiotic detoxification in liver. Similar liver expression of UGT2Es were also observed in rabbits and pigs [10,62], and genetically rodents and human might not have equivalent UGT2Es [10,11]. Although detailed substrate specificity of these isoforms was not yet clarified, and the importance of these isoforms in liver detoxification are still vague. However, there are possible isoforms affect UGT liver activity differences, since we detected variation of these isoforms among Carnivora.

### 4.4. Limitation of This Study

In this analysis, we utilized the genome assembly data for multiple Carnivorans, which has several limitations. Genome assemble data only represent one individual type of genomic information and several variants could happen in all genomic assembly. Additionally, genome assembly qualities should be considered. Although recent genomic assemble quality become improved with log-read sequencing techniques, and several chromosomal-level assemblies were available for even wildlife Carnivorans, some assemblies have relatively low quality, which made it difficult to analyze some of the UGT genes. 

Another limitation is the annotation quality. The analysis in this study highly relied on the annotation data of NCBI database. The NCBI Eukaryotic Genome Annotation Pipeline [63] was based on mapping of transcript data in each species and in silico prediction, and some species lacked the liver transcriptome data. Since major UGT isoforms are highly expressed in the liver, annotation error could happen in some species. For avoiding annotation error, we used BLAST to align the possible UGT isoforms on each genome to reduce the chance of annotation error, and there could still be some possible annotation error. 

Although further detailed analysis was highly important, we considered that this genomics analysis has high importance to evaluate wildlife UGT genome in comprehensive Carnivora taxa. 

## 5. Conclusions

This study, for the first time, revealed the evolutionary characteristics of UGT in several Carnivora species, providing a more comprehensive understanding of UGT duplication and loss event in this clade. Our results indicate that omnivorous species such as canids and some bears might have been subjected to significant selective pressure on both the UGT1A and 2B subfamily. Furthermore, we found significant contraction of UGT genes in pinnipeds and Felidae, providing additional indications that limited genetic variation of UGTs in this group is much more comprehensive than previously assumed. Although genomic information for some species still requires improved annotation or assembly, our findings provide fundamental information for more accurate extrapolation of pharmacokinetic or toxicokinetic result from experimental animals to wild carnivorans which are daily exposed to numerous anthropogenic chemicals.

## Figures and Tables

**Figure 1 animals-12-02954-f001:**
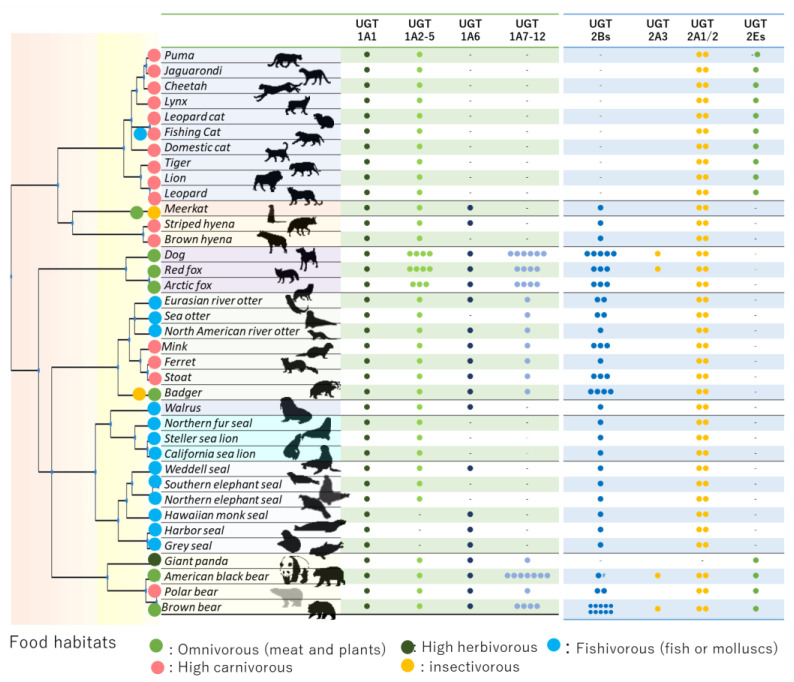
Gene numbers for UGT1A, 2A, 2B, and 2C/E are shown by number of small filled circles. Large filled circles next to the scientific name of each species are colored by known diet. In brown bear (Ursus arctos), we detected three coding loci and all isoforms except “partial” or “low quality” were counted in this case. In black bear, we omitted six other partially coded genes. The phylogenetic tree was created with TimeTree 5 [23]. Manually curated unannotated isoforms, in badger, ferret, and sea otter, respectively, were also counted in this table.

**Figure 2 animals-12-02954-f002:**
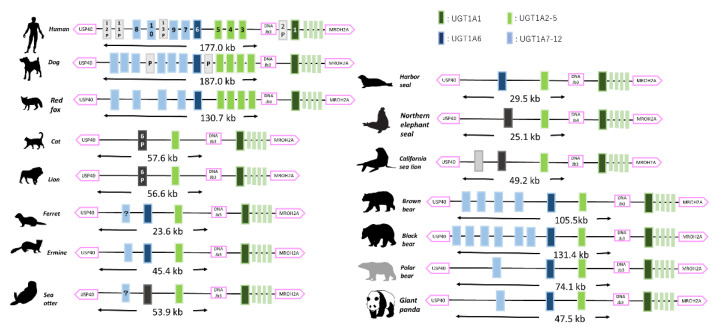
Synteny analysis of UGT1As in Carnivora. Synteny of UGT1A coding loci among Carnivora is shown. Representative species for each family were selected. UGT1As are known to share common exons (2–5) among isoforms and are shown as pale green blocks. UGT1A1 is dark green, UGT1A2–5 is bright green, UGT1A6 is dark blue, and UGT1A7–12 is pale blue. Pseudogenes are shown as black or gray blocks. Lengths of coding loci from DNAJB3 to MROH2A are also shown as indicator for genetic loss in these loci. Manually curated unannotated isoforms are shown with “?”.

**Figure 3 animals-12-02954-f003:**
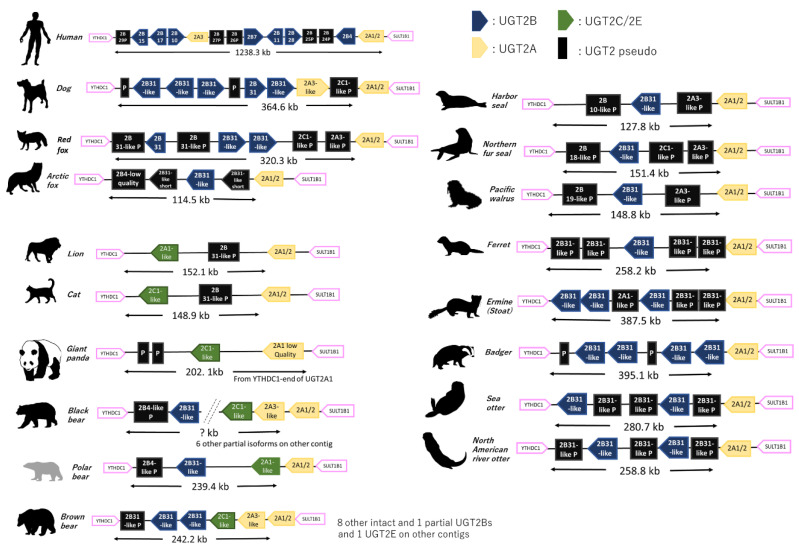
Synteny analysis of UGT2s in Carnivora. Synteny of UGT2 coding loci among Carnivora is shown. Representative species for each family were selected. UGT2A1/2 are known to share common exons (2–5) among isoforms and are shown as yellow blocks with the same color in other UGT2As. UGT2Bs are navy, UGT2E/2Cs are green, and pseudogenes are black. Lengths of coding loci from UGT2As to YTHDC1 are also shown as indicators for genetic loss in these loci.

**Figure 4 animals-12-02954-f004:**
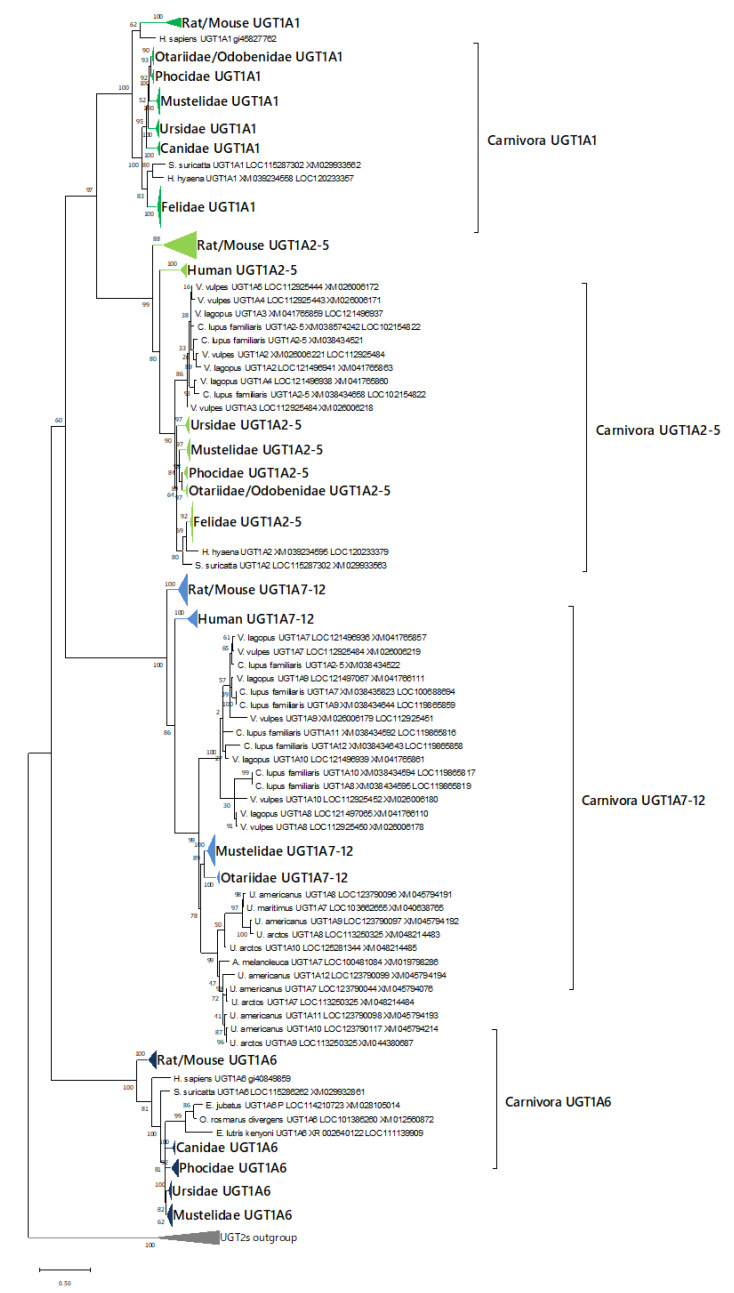
Phylogenetic tree of UGT1As. Phylogenetic tree of UGT1A sequences in human, mouse, rat, and carnivorans. Gene sequences of protein-coding regions for each isozyme were analyzed. The numbers next to the branches indicate the number of occurrences per 100 bootstrap replicates. Genes and clades are tentatively labeled with carnivoran UGTs examined in this article. Clades of rodents, and human UGT1A1, UGT1A2–5, UGT1A6, and UGT1A7–12 in the phylogenetic tree are shown as triangles of the following colors: dark green for UGT1A1s, bright green for UGT1A2–5s, deep blue for UGT1A6s, and pale blue for UGT1A7–12s. UGT2As are shown as an outgroup.

**Figure 5 animals-12-02954-f005:**
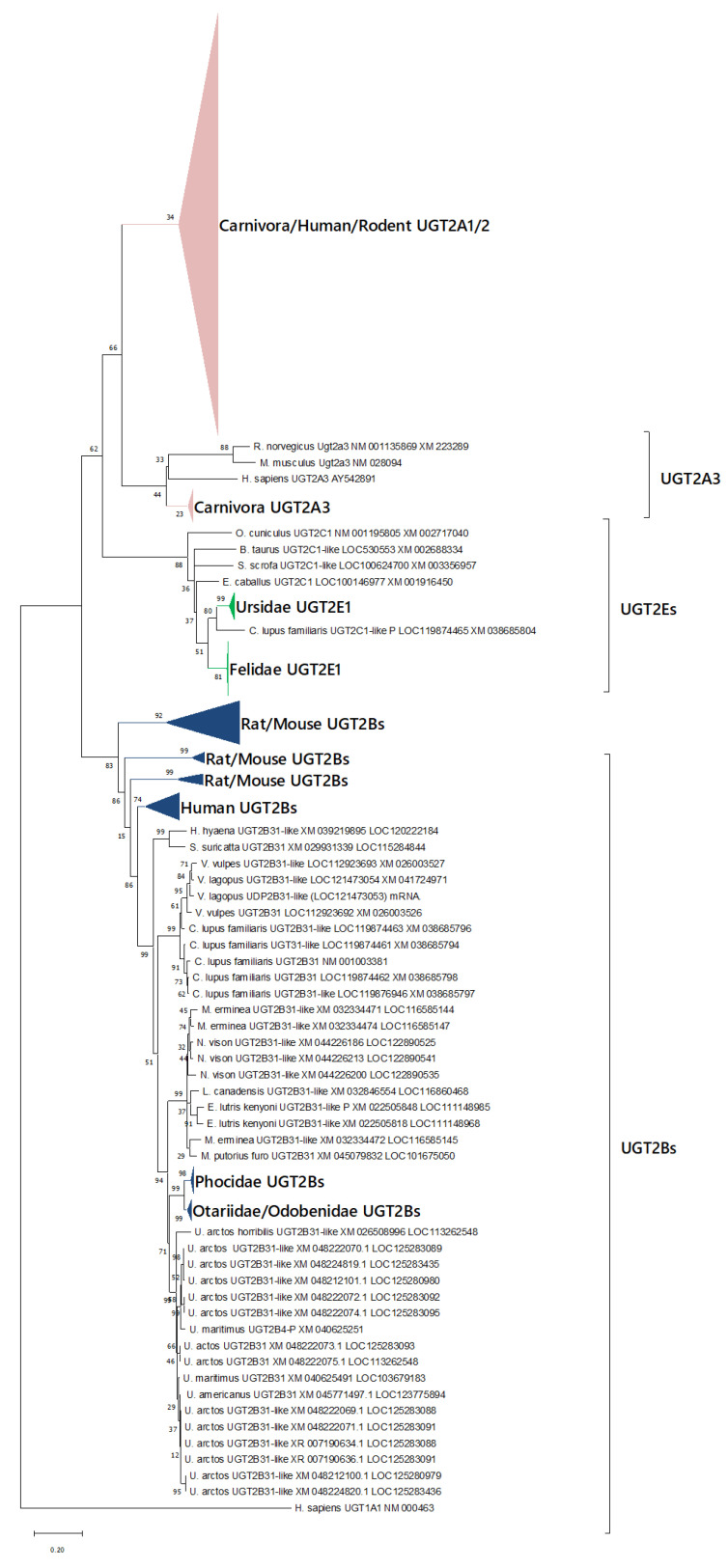
Phylogenetic tree of UGT2s. Phylogenetic tree of UGT2s sequences in human, mouse, rat, and carnivorans. Gene sequences of protein-coding regions for each isozyme were analyzed. The numbers next to the branches indicate the number of occurrences per 100 bootstrap replicates. Genes and clades are tentatively labeled with carnivoran UGTs examined in this article. Clades of rodent and human UGT2As, UGT2Bs, and UGT2Es in the phylogenetic tree are shown as triangles of the following colors: navy for UGT2Bs, green for UGT2Es, and pink for UGT2As. UGT1As are shown as an outgroup.

## Data Availability

Sequence data we retrieved are available in Appendix A: UGT1A_Carnivora_nucleotide.fas, UGT1A_Carnivora_protein.fas, UGT2s_Carnivora_nucleotide.fas, UGT2s_Carnivora_protein.fas.

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
