# Peer review of "Duplication, Loss, and Evolutionary Features of Specific UDP-Glucuronosyltransferase Genes in Carnivora (Mammalia, Laurasiatheria)"

_animals, 2022, doi:10.3390/ani12212954_

Round 1
Reviewer 1 Report
This is a very interesting study of the diversity and genomic phylogeny of UGT detoxification among Carnivora. The overall contention is that diet is a driving force for expansion and contraction of detoxification UGTs. The main weakness of the study is that it relies heavily on the quality and annotation of genomes for the species analyzed. This limitation needs to be made clearer in the manuscript. Regardless, the currently available data seems to provide a good starting point for these investigations. The other weakness is the inclusion of the UGT2A genes in the analysis. The UGT2As are primarily involved in olfaction (odorant termination) without clear evidence for a role in dietary detoxification (perhaps anteaters??). Regardless - it could be argued the work would stand alone without the UGT2As - which could be a separate future study. Other specific comments that need to be addressed are as follows:
1. Lines 110-113: Genome annotation can vary greatly in quality depending in part on how the identity of coding regions are validated - such as through use of transcriptome data. This is particularly problemmatic for highly homologous genes - and also differentially spliced genes - such as the UGT1A gene. Sometimes only manual annotation and experimental confirmation of cDNA expression is required. How was it ascertained that all of the UGT coding regions (especially the UGT1A exons 1) had been identified in each species?
2. Lines 110-113: Like UGT1A, UGT2A1/2 is a single gene with differentially spliced first exon in mammalian species studies so far. Only the dog UGT2A1 sequence (not the UGT2A2 sequence) was used for this search. Why weren’t the first exons used instead - like UGT1A??
3. Lines 110-113: UGT2A3 forms a separate gene (and clade - Figure 5) from UGT2A1/2. Why was UGT2A3 not included??
4. Line 159: Did the measurement of the length of the coding locus take into account the presence of any gaps in the sequence in those regions?? Clusters of homologous genes can be difficult to sequence gap-free without using long read approaches (PacBio).
5. The domestic dog (Canis lupus familiaris) is a subspecies of grey wolf (Canis lupus). There is evidence for genomic adaptation related to diet (carnivore to omnivore) of wolf to domestic dog (PMID: 23354050). Were possible differences in the UGT genes between dog and wolf considered or evaluated??
6. Brown bear 2B expansion Fig 3. The Brown bear syteny anlysis shows 2 additional contigs without any synteny support (flanking genes). If these are simply short contigs - it is possible that these are artifacts (7 x 2B31 repeat) or partial sequences of the main (larger) contig used as reference. Including these is unnecessary and distracting. Suggest removing those 2 additional contigs from the Brown bear map and just putting a note underneath - like done with the Black bear map.
7. Lines 290-292 These statements regarding the brown and black bear vs polar bear needs to be tempered by the previous point in that the data regarding UGT2B expansion is preliminary and would need to be confirmed by other sequencing methods (long read?).
8. UGT official names. Any full length cDNAs from this analysis - derived from mRNA sequencing - including Rnaseq - but not gene prediction software - should be submitted to the UGT nomenclature committee to obtain official names.
9. Figure 1 - UGT2A3 and UGT2A1/2s seem to be lumped together in the gene count. However, UGT2A3 genes seem to form a separate clade from UGT2A1/2 (Fig 5). Why?
10. Figure 5: Human UGT2A1 clusters with most of the UGT2A2 genes in other species, while human UGT2A2 clusters with UGT2A1 in most other species. Why? Was the full-length amino acid sequence used - or only exon 1 region?? Since UGT2A1/2 are spliced from the same gene would it be better to only use exon 1??
11. Figure 5: Why were mouse/rat/human UGT1As (fig. 4) and UGT2Bs used for phylogeny analysis - but not mouse/rat UGT2As or human UGT2A3 used??
12. Results and discussion text do not describe UGT2A or UGT2C/2E results. Consider removing UGT2A/2C/2E? Or at least collapse them??
13. The Giant Panda seems to have very restricted UGT isoform diversity - only 4 UGT1As, no UGT2Bs and one UGT2C/E (Fig 1 and 2). Even less than the Polar bear (carnivore diet). However Panda species is classified as highly herbivorous which would argue for expansion, rather than contraction relative to other Carnivora. Why?? Giant Panda have a diet restricted to bamboo - so is it possible that bamboo contains low amounts of substances requiring glucuronidation?? This would imply that it is not just herbivory - but the type of herbivory that is important. Discussion of this is warranted given the phylogeny of panda within Carnivora.
Author Response
Reviewer: 1
The first, we sincerely thank you for taking the time to review our paper. I am attaching my response to your point below, as I think it is easier to read in WORD.
This is a very interesting study of the diversity and genomic phylogeny of UGT detoxification among Carnivora. The overall contention is that diet is a driving force for expansion and contraction of detoxification UGTs. The main weakness of the study is that it relies heavily on the quality and annotation of genomes for the species analyzed. This limitation needs to be made clearer in the manuscript. Regardless, the currently available data seems to provide a good starting point for these investigations.
We appreciate your critical review regarding our manuscript.
About the limitation of assemble quality and annotation data, we also consider this point as important issues. Thus, we added the section in Discussion, mentioning the limitation of the data,
Discussion L420-437:
Limitation of this study
In this analysis, we utilized the genome assembly data for multiple Carnivorans, and still these data analysis has several limitations. Genome assemble data only represents one individual genomic information and several variants could happen in all genomic assembly. Also, genome assembly qualities should be considered. Although recent genomic assemble quality become improved with log-read sequencing techniques, and several chromosomal-level assemblies were available for even wildlife Carnivorans, some assembly have relatively low quality which made us difficult to analyze some of UGT genes.
Another limitation is on the annotation quality. The analysis in this study highly relied on the annotation data of NCBI database. The NCBI Eukaryotic Genome Annotation Pipeline [63] were based on mapping of transcript data in each species and in silico prediction, and some species lacked the liver transcriptome data. Since major UGT isoforms highly expressed in the liver, annotation error could happen in some species. For avoiding annotation error, we used BLAST to align the possible UGT isoforms on each genome to reduce the chance of annotation error, still there could be some possible annotation error.
Although further detailed analysis was highly important, we considered this genomic data analysis have strong importance to evaluate wildlife UGT genome in comprehensive Carnivora taxa.
The other weakness is the inclusion of the UGT2A genes in the analysis. The UGT2As are primarily involved in olfaction (odorant termination) without clear evidence for a role in dietary detoxification (perhaps anteaters??). Regardless - it could be argued the work would stand alone without the UGT2As - which could be a separate future study.
We deeply appreciate this comment. For UGT2A1/2s, the expression has been detected mainly in olfactory epithelium and not in liver mainly. However, UGT2A3, and 2Es have been reported as high expression in liver in several mammals. In this study about Carnivora, we also detected the slight differences in UGT2Es and UGT2A3, and we consider these points also important to be assessed. That’s why we added the UGT2As and 2Es in our analysis. We added the following section in Discussion mentioning about possible importance of UGT2As and 2Es.
L410-418
We also detected slight differences of UGT2A3s and 2Es among Carnivora. We found the existence of UGT2A3 in some Canids and Ursid, and UGT2Es in Felids and Ursids. UGT2A1 and 2A2s have been recognized as non-liver expressed isoforms as mentioned above [51,52]. However, UGT2A3 were detected in liver of some primates and rodents [51,52,61], suggesting possible role for xenobiotic detoxification in liver. Similar liver ex-pression of UGT2Es were also observed in rabbits and also pig [10,62], and genetically rodents and human might not have equivalent UGT2Es [10,11]. Although detailed substrate specificity of these isoforms was not yet clarified and still the importance of these isoforms in liver detoxification are vague. However, there are possible isoforms affect UGT liver activity differences, since we detected variation of these isoforms among Carnivora.
Other specific comments that need to be addressed are as follows:
1. Lines 110-113: Genome annotation can vary greatly in quality depending in part on how the identity of coding regions are validated - such as through use of transcriptome data. This is particularly problematic for highly homologous genes - and also differentially spliced genes - such as the UGT1A gene. Sometimes only manual annotation and experimental confirmation of cDNA expression is required. How was it ascertained that all of the UGT coding regions (especially the UGT1A exons 1) had been identified in each species?
Thank you for your comments regarding the UGT genes annotation.
We also considered that genome annotation can vary greatly, and we further crated the annotation using BLAST on genome assembly sequence to reduce the annotation error. Through this further curation, we found only one possible isoforms of UGT1A7-12 in some Mustelidae (Badger, ferrets, and sea otter). In other carnivora, we didn’t find any other possible UGTs. From these circumstances, we changed the Fig. 1 and Fig. 2 into newly curated version. Please check the renewed version.
Also, we added the sentence followed, mentioning about this in Materials and methods,
L112-115:
We also used BLAST to align query UGT sequences onto genome assembly sequences (RefSeq genome Database (refseq_genomes), optimized for somewhat similar sequences) to make sure we did not miss other possible UGT isoforms on each species.
Moreover, in Result, we added the sentences in L168-171.
Further BLAST search onto each genome sequence revealed other possible un-annotated UGT1A7-12 isoforms and added on Figure 1, Figure 2 and further phylogenetic analysis. Manually curated un-annotated isoforms were shown with “?” on the block in Figure 2.
Also in the Figure 1 legend, L180-181
Manually curated one un-annotated isoforms, in badger, ferret and sea otter respectively, were also counted in this table.
Also in the Figure 2 legend, L189-190
Manually curated un-annotated isoforms were shown with “?” on the block.
However, we consider these changes were still minor, and the previous Discussion could be applicable for this issue. We again appreciate your critical comment.
Lines 110-113: Like UGT1A, UGT2A1/2 is a single gene with differentially spliced first exon in mammalian species studies so far. Only the dog UGT2A1 sequence (not the UGT2A2 sequence) was used for this search. Why weren’t the first exons used instead - like UGT1A??
Thank you for asking about this point. In this analysis, we added whole UGT2A1/2s because we had to know clear phylogenetical differences between UGT2A3, 2Es and 2Bs to make sure NCBI annotation as UGT2A1/2 were correct, because NCBI annotation and naming system sometimes made mistake. Since other UGT2s were not coded as single gene with differentially spliced first exon, we added whole sequences of UGT2A1/2s in this analysis. Thank you again for your comment.
Lines 110-113: UGT2A3 forms a separate gene (and clade - Figure 5) from UGT2A1/2. Why was UGT2A3 not included??
That was our mistake. We apologize for that point, and thank you for mentioning. We added UGT2A1/2 and 2A3 in L111.
Line 159: Did the measurement of the length of the coding locus take into account the presence of any gaps in the sequence in those regions?? Clusters of homologous genes can be difficult to sequence gap-free without using long read approaches (PacBio).
Thank you for your comment. We made sure the UGT coding regions had no gaps on the assemble in Fig. 2 and Fig. 3. However, like your comment, we did find some gaps in several animals (e.g., UGT1A loci in weddell seal and pacific walrus, maybe because the assembles are old and only short reads were utilized.). In the Figure 2 and 3, we omitted these species. Thank you again for your comments.
The domestic dog (Canis lupus familiaris) is a subspecies of grey wolf (Canis lupus). There is evidence for genomic adaptation related to diet (carnivore to omnivore) of wolf to domestic dog (PMID: 23354050). Were possible differences in the UGT genes between dog and wolf considered or evaluated??
We also considered this point important. We added the followed sentences in the Discussion, L321-327
Our phylogenetic analysis of UGT2Bs suggested dog and foxes had totally different evolutional expansions. That suggest UGT2Bs expansion in each species occurred after the divergence of these species. Previous report provided the evidence that novel genomic adaptations of modern dogs to thrive on a diet rich in starch [45]. Together with our phylogenetic analysis, these reports suggested UGT2Bs expansion might be specific to dogs, and not in wolves. Further genomic comparison between these species should be taken into consideration in future study.
Brown bear 2B expansion Fig 3. The Brown bear synteny analysis shows 2 additional contigs without any synteny support (flanking genes). If these are simply short contigs - it is possible that these are artifacts (7 x 2B31 repeat) or partial sequences of the main (larger) contig used as reference. Including these is unnecessary and distracting. Suggest removing those 2 additional contigs from the Brown bear map and just putting a note underneath - like done with the Black bear map.
We appreciate your suggestion and we changed the Fig. 3.
Lines 290-292 These statements regarding the brown and black bear vs polar bear needs to be tempered by the previous point in that the data regarding UGT2B expansion is preliminary and would need to be confirmed by other sequencing methods (long read?).
We appreciated and added the sentence below in Discussion.
L302-305:
Although these data of UGT2Bs expansions were still preliminary and further study to make sure these expansion and isoform number differences in these bears are needed, these data suggested the possible variation of UGT2Bs within Ursids.
UGT official names. Any full length cDNAs from this analysis - derived from mRNA sequencing - including Rnaseq - but not gene prediction software - should be submitted to the UGT nomenclature committee to obtain official names.
Thank you very much for that comment. In this analysis, we didn’t conduct Rnaseq analysis or further conventional sequencing analysis ourselves, and we have no data to submit for UGT nomenclature committee. Thank you again for your comment.
Figure 1 - UGT2A3 and UGT2A1/2s seem to be lumped together in the gene count. However, UGT2A3 genes seem to form a separate clade from UGT2A1/2 (Fig 5). Why?
We are grateful to your comment, and we separated the UGT2A1/2 and UGT2A3 in Fig. 1.
Figure 5: Human UGT2A1 clusters with most of the UGT2A2 genes in other species, while human UGT2A2 clusters with UGT2A1 in most other species. Why? Was the full-length amino acid sequence used - or only exon 1 region?? Since UGT2A1/2 are spliced from the same gene would it be better to only use exon 1??
That was our mistake. We do apologize for the point and modified the Fig.5.
About using exon 1, as we mentioned above, we added whole sequences of UGT2A1/2s because we had to know clear phylogenetical differences between UGT2A3, 2Es and 2Bs to make sure NCBI annotation as UGT2A1/2 were correct. Thank you again for your comment.
Figure 5: Why were mouse/rat/human UGT1As (fig. 4) and UGT2Bs used for phylogeny analysis - but not mouse/rat UGT2As or human UGT2A3 used??
We appreciate that comment and re-analyzed the phylogeny of UGT2s adding rodent UGT2A1/2 and human 2A3, and changed the Fig. 5.
Results and discussion text do not describe UGT2A or UGT2C/2E results. Consider removing UGT2A/2C/2E? Or at least collapse them??
We appreciate the comment. We still consider the importance of UGT2A3s and UGT2Es in Carnivora, mentioned previously. However, as you pointed, UGT2A1/2 were not discussed in detail in our manuscript. Therefore, we collapsed UGT2A1/2 into one clade on Fig. 5. Thank you very much.
The Giant Panda seems to have very restricted UGT isoform diversity - only 4 UGT1As, no UGT2Bs and one UGT2C/E (Fig 1 and 2). Even less than the Polar bear (carnivore diet). However Panda species is classified as highly herbivorous which would argue for expansion, rather than contraction relative to other Carnivora. Why?? Giant Panda have a diet restricted to bamboo - so is it possible that bamboo contains low amounts of substances requiring glucuronidation?? This would imply that it is not just herbivory - but the type of herbivory that is important. Discussion of this is warranted given the phylogeny of panda within Carnivora.
Thank you very much for your comment. We have already discussed in Discussion L309-317 that giant pandas have bamboo-specialist diet feature and that might be partial reason for contracted UGT genes. Actually, bamboo might contain variety of bioactive flavonoids and flavonoids mainly metabolized by glucuronidation.
Still, we need to consider about the alternative pathways, like other Phase II metabolism enzymes. Therefore, we added the sentence below in L. We do appreciate your comments.
L 318-320
Further investigation about alternative pathways (e.g., sulfotransferase or other phase II metabolism enzymes) in giant panda to deal with these bamboo-derived phytochemicals or flavonoids need to be assessed in the future.
We again deeply appreciate your comprehensive and critical review thorough entire our manuscript. We would also like to attach the revised version of Figures (same as revised manuscript) here. Could you please kindly check these figures.
Reviewer 2 Report
In this study, the author gave a detailed analysis of the gene synteny of UGTs, especially about UGT1As, 2s in Carnivora. In result part, the authors found UGT1A duplication in Canidae, brown bear and black bear, and UGT2Bs duplication in Canidae, some Mustelidae, and Ursidae. In addition, they also observed gene contraction or loss of UGT1As in Phocidae, Otariidae, Felidae, Giant panda and polar bear, gene loss of UGT2B in Felidae and Giant panda. As authors explained, the evolution process of UGTs is related to diet of different animals, which agrees with previously reported results. The authors also noticed the evolutionary characteristics of UGTs is complex and species specific, and close related species may have different UGT gene counts. Diet is only a part of natural history and can not completely explain all phenomna. Fox example, Badger is a generalist and Omnivore, which have similar number of UGT2Bs with dog and fox in Fig.3, but for UGT1A in fig.1, duplication is no found.
I suggest authors had better list complete sequence data and alignment data for UGT1A and UGT2Bs in supplemental data part.
Author Response
Reviewer: 2
We deeply appreciate your review of our paper. I am attaching below my response to your remarks.
In this study, the author gave a detailed analysis of the gene synteny of UGTs, especially about UGT1As, 2s in Carnivora. In result part, the authors found UGT1A duplication in Canidae, brown bear and black bear, and UGT2Bs duplication in Canidae, some Mustelidae, and Ursidae. In addition, they also observed gene contraction or loss of UGT1As in Phocidae, Otariidae, Felidae, Giant panda and polar bear, gene loss of UGT2B in Felidae and Giant panda. As authors explained, the evolution process of UGTs is related to diet of different animals, which agrees with previously reported results. The authors also noticed the evolutionary characteristics of UGTs is complex and species specific, and close related species may have different UGT gene counts. Diet is only a part of natural history and cannot completely explain all phenomena. Fox example, Badger is a generalist and Omnivore, which have similar number of UGT2Bs with dog and fox in Fig.3, but for UGT1A in fig.1, duplication is no found.
I suggest authors had better list complete sequence data and alignment data for UGT1A and UGT2Bs in supplemental data part.
We appreciate your suggestion and we uploaded the protein sequence data file as FASTA format of UGT1As and 2s as Supplemental Data. We also modified the sequence list as table format in Supplemental Data.
We again grateful to your kind and critical review.

Round 2
Reviewer 1 Report
Thank you for your responses which have addressed all on my concerns except one. The remaining concern for me is the discussion of the panda contraction in UGT diversity (UGT2B loss) relative to other bear species.
My thinking is that a diet restricted to bamboo (dietary herbivore specialist) should also be restricted in the diversity of phytochemicals. Consequently only a limited number of types of UGTs would be required to detoxify the diet. Although there could be very high amounts of those UGT enzymes. On the other hand black and brown bears have a more generalist diet containing many different plant species (berries, roots, etc etc) should have a much higher diversity of UGT types to deal with those phytochemicals.
Line 311 "The cause for this might be the species’ exclusive bamboo diet, which may have led to the evolution of settled isoforms to deal with bamboo phytochemicals such as flavonoids [38].”
I don’t understand what is mean’t by “settled” in this context. Also - I cannot access the reference. It is likely that bamboo have phytochemicals (secondary plant metabolites). However what needs to be known is the diversity of these metabolites. And whether different UGTs would be required to metabolize them.
Line 313 “Some reports have supported this interpretation, showing that specialists tend to have a wider variety of phytochemical metabolites than generalists [39–41].”
Firestly, I don’t agree that these reports support the prior interpretation.
Paper #39 is a review paper that deals with the “detoxification limitation hypothesis” - which posits that when an animal eats a plant with one type of metabolite, it will eat that until a tolerable threshold is achieved and then eat a plant with a different metabolite profile. This seems to explain how an animal with a particular detoxification profile deals with plant toxins by adjusting what they eat. This is an explanation for an immediate phenomenon. Rather what you are dealing with above is how the UGT system responds to changes in diet over evolutionary time.
Paper #40 explores this hypothesis in the context of specialist vs generalist herbivores. They show lower exposure to a particular plant chemical by a specialist (from a plant they exclusively consume) versus a generalist. This makes sense since the specialist has adapted to eat that particular plant toxin. However it doesn’t address whether the specialist has less diversity of toxin handling mechanisms - just that it has a higher capacity.
Paper#41 addresses whether specialists differ from generalists in the degree of phase 1 versus phase 2 metabolism - which was not explored in this paper.
I don’t find any evidence in these papers that “specialists tend to have a wider variety of phytochemical metabolites than generalists”. Shouldn’t the opposite be true??
Please clarify these 2 sentences.
Author Response
Reviewer: 1
Thank you for your responses which have addressed all on my concerns except one. The remaining concern for me is the discussion of the panda contraction in UGT diversity (UGT2B loss) relative to other bear species.
My thinking is that a diet restricted to bamboo (dietary herbivore specialist) should also be restricted in the diversity of phytochemicals. Consequently only a limited number of types of UGTs would be required to detoxify the diet. Although there could be very high amounts of those UGT enzymes. On the other hand black and brown bears have a more generalist diet containing many different plant species (berries, roots, etc etc) should have a much higher diversity of UGT types to deal with those phytochemicals.
Line 311 "The cause for this might be the species’ exclusive bamboo diet, which may have led to the evolution of settled isoforms to deal with bamboo phytochemicals such as flavonoids [38].”
I don’t understand what is mean’t by “settled” in this context. Also - I cannot access the reference. It is likely that bamboo have phytochemicals (secondary plant metabolites). However what needs to be known is the diversity of these metabolites. And whether different UGTs would be required to metabolize them.
We do apologize for making you confused, and the sentence seems very misunderstanding.
We meant in this “settled isoforms” is that,
“UGT isoforms which adapted or evolved to deal with bamboo-specific phytochemicals.” Since we hypothesized that lower number of UGTs can deal with bamboo-specific chemical metabolism, only “settled isoforms” might be needed for giant panda.
Obviously, as your suggestion, this sentence was totally unclear, so we changed the sentence as follows. We deeply appreciate your very critical and precise comment.
L311-319
The cause for this might be the species’ exclusive bamboo diet, which may have led to the evolution of settled isoforms to deal with bamboo phytochemicals such as flavonoids
↓
The cause for this might be the species’ exclusive bamboo diet. Since strictly limited variety of diet the giant panda take, they might take much narrower set of phytochemicals that other bears intake. These might lead them to have only limited number of UGT isoforms. Although bamboo itself also contains certain variety of phytochemicals which may be metabolized by UGTs [38], the lower number of UGTs (which may be specialized for bamboo-phytochemical metabolism) might be enough for them to deal with their dairy intake of the chemicals.
Also for Reference [38], we modified the information in L574-575. Sorry for the inconvenience.
Line 313 “Some reports have supported this interpretation, showing that specialists tend to have a wider variety of phytochemical metabolites than generalists [39–41].”
Firstly, I don’t agree that these reports support the prior interpretation.
Paper #39 is a review paper that deals with the “detoxification limitation hypothesis” - which posits that when an animal eats a plant with one type of metabolite, it will eat that until a tolerable threshold is achieved and then eat a plant with a different metabolite profile. This seems to explain how an animal with a particular detoxification profile deals with plant toxins by adjusting what they eat. This is an explanation for an immediate phenomenon. Rather what you are dealing with above is how the UGT system responds to changes in diet over evolutionary time.
Paper #40 explores this hypothesis in the context of specialist vs generalist herbivores. They show lower exposure to a particular plant chemical by a specialist (from a plant they exclusively consume) versus a generalist. This makes sense since the specialist has adapted to eat that particular plant toxin. However it doesn’t address whether the specialist has less diversity of toxin handling mechanisms - just that it has a higher capacity.
Paper#41 addresses whether specialists differ from generalists in the degree of phase 1 versus phase 2 metabolism - which was not explored in this paper.
I don’t find any evidence in these papers that “specialists tend to have a wider variety of phytochemical metabolites than generalists”. Shouldn’t the opposite be true??
We deeply appreciate your critical comment. As your mentioned, this sentence was incorrect. Therefore, we changed the sentence as follows. And also references [39-41] were also changed. Hope that makes it much better
L317-324
However, in mammals, some reports have shown the opposite traits which, specialist tend to have a wider variety of xenobiotic metabolism enzymes than generalists. In koala, to deal with their daily eucalypt-specialized diet, their genome seemed to expand CYP2C subfamilies [39]. Also, juniper-specialist woodrat and red tree voles, which have Douglas-fir specialized diet, observed higher gene copy number of CYPs, compared to closely related generalist species [40,41]. Although these reports have been mainly discussed about CYP genes, which were not addressed in study.
Also, we modified the references below.
- Johnson, R.N.; O’Meally, D.; Chen, Z.; Etherington, G.J.; Ho, S.Y.W.; Nash, W.J.; Grueber, C.E.; Cheng, Y.; Whittington, C.M.; Dennison, S.; et al. Adaptation and Conservation Insights from the Koala Genome. Nat. Genet. 2018, 50, 1102–1111, doi:10.1038/s41588-018-0153-5.
- Kitanovic, S.; Marks-Fife, C.A.; Parkes, Q.A.; Wilderman, P.R.; Halpert, J.R.; Dearing, M.D. Cytochrome P450 2B Diversity in a Dietary Specialist-the Red Tree Vole (Arborimus Longicaudus). J. Mammal. 2018, 99, 578–585, doi:10.1093/jmammal/gyy039.
- Kitanovic, S.; Orr, T.J.; Spalink, D.; Cocke, G.B.; Schramm, K.; Wilderman, P.R.; Halpert, J.R.; Dearing, M.D. Role of Cytochrome P450 2B Sequence Variation and Gene Copy Number in Facilitating Dietary Specialization in Mammalian Herbivores. Mol. Ecol. 2018, 27, 723–736.
We again deeply appreciate your precise, detailed, critical and kind reviews through entire our manuscript.